# Challenges and Strategies in Conducting Population Health Research during the COVID-19 Pandemic: Experience from a Nationwide Mixed-Methods Study in Bangladesh

**DOI:** 10.3390/ijerph20095629

**Published:** 2023-04-25

**Authors:** Ashraful Kabir, Md Nazmul Karim, Jahirul Karim, Baki Billah

**Affiliations:** 1Department of Epidemiology and Preventive Medicine, School of Public Health and Preventive Medicine, Monash University, Melbourne, VIC 3004, Australia; 2Directorate General of Health Services, Ministry of Health and Family Welfare, Dhaka 1212, Bangladesh

**Keywords:** Bangladesh, COVID-19 pandemic, fieldwork, mixed-methods study, population health research

## Abstract

**Introduction:** Globally, the coronavirus (COVID-19) pandemic poses fundamental challenges in everyday life. Various controlling measures, including nationwide lockdowns, movement restrictions, travel bans, social distancing, and improved hygiene practices, have been widely introduced to curtail transmission of the disease. Notably, these measures have affected the execution of population health research that typically involves face-to-face data collection. This paper details a subjective reflective account of the challenges and mitigating strategies in conducting a nationwide study during the COVID-19 pandemic in 2021. **Challenges and strategies:** The research team faced a wide range of challenges in conducting this study. The major categories of challenges were defined as follows: (i) challenges relating to the COVID-19 pandemic, such as insufficient access to field sites; (ii) challenges related to contextual factors, such as cultural and gender sensitivity and extreme weather events; and (iii) challenges related to data quality and validity. The key mitigating strategies to overcoming these challenges included engaging a local-level field supervisor, hiring data collectors from respective study sites, incorporating team members’ reviews of literature and experts’ views to develop research instruments, modifying original research instruments, organizing regular meetings and debriefing, adjusting field operation plans, building gender-sensitive teams, understanding local norms and adopting culturally appropriate dress codes, and conducting interviews in local languages. **Conclusions:** This paper concludes that despite several COVID-19-related challenges coupled with contextual factors, data were successfully collected through timely and successful adaptations of several mitigating strategies. The strategies adopted in this study may be useful for overcoming unforeseeable challenges in planning and conducting future population-based health research in similar circumstances elsewhere.

## 1. Introduction

The coronavirus (COVID-19) pandemic has brought unprecedented change to everyday life worldwide [1,2]. As of 29 January 2023, COVID-19 has infected over 753 million people and caused over 6.8 million deaths around the world [3]. Since the declaration of the COVID-19 pandemic in March 2020, various controlling measures such as nationwide lockdowns, restricted movement, travel bans, social distancing, and improved hygiene practices have been widely introduced [4]. Despite these measures, the ongoing number of COVID-19 infection cases remains a concern, and a resurgence in COVID-19 cases is being recorded in many parts of the world [5,6,7]. As it has many aspects of life today, the emergence of COVID-19 has notably affected research activities [8,9,10], especially population research, which has been heavily disrupted during the COVID-19 pandemic [10,11]. Population health research involves collecting primary data from individuals and/or at the community level. The primary data are typically collected through face-to-face interviews known as fieldwork [9,11]. The degree of challenges in the execution of studies varies depending on the method, the design, and the context under which the respective studies are conducted [12]. Studies that employ mixed methods and/or qualitative designs are more likely to be impacted by the COVID-19 pandemic compared with those that use survey methods due to the need of the former for considerable human interaction [9].

Due to the pace of the pandemic, many institutions and individuals in resource-affluent countries sharply adopted internet-based hands-off modes, using telephone technology to execute fieldwork as an alternative to face-to-face interactions [13,14,15]. The use of updated digitalized technology has become popular and is effective at minimizing the fieldwork hurdles associated with the COVID-19 pandemic. Nevertheless, the scope for using technology-based hands-off modes is restricted and unsuitable in places where technological facilities are limited [15]. The availability and accessibility of internet-based technology and users’ comfortability and willingness to engage with it are essential factors in adapting these modes successfully [14,15]. More importantly, the feasibility of these adaptations relies on the design and methodology of the research, as well as contextual factors such as sociodemographic features, geographic location, sociocultural characteristics, and traditions of the studied community [16,17]. Therefore, context-oriented, practical, and actionable remedial strategies are appropriate to overcome the challenges associated with the design and data collection methods of the study.

In Bangladesh, as in many resource-poor countries, the execution of population-based research during the COVID-19 pandemic remains notably challenging. The scope for using the above-mentioned alternative strategies is noticeably limited due to the lack of technological resources, poor accessibility in outreach areas, the associated cost, and the willingness and comfortability of the respective users. Specifically, the application of these alternative modes is most likely inappropriate and unsuitable for conducting population-based health research in rural settings [18]. Furthermore, there is a set of context-specific hurdles that affect implementing fieldwork in a smooth and timely fashion and gathering quality and valid data [18,19]. In the context of Bangladesh, the existing literature presents a shortage of subjective reflective accounts on challenges and mitigating strategies in conducting population health research during the pandemic. To the best of our knowledge, a few studies reported challenges and/or strategies concerning population-based research in Bangladesh during pre-pandemic times [12,19,20,21]. However, a detailed study account of the challenges faced by researchers conducting nationwide population health research during the COVID-19 pandemic and the mitigating strategies used to overcome them has not been reported. Against this backdrop, this paper presents a subjective reflective account of the challenges and the mitigating strategies used to overcome them by conducting a population-based nationwide mixed-methods health study during a full-blown COVID-19 pandemic. Therefore, the experiences reported in this study may prove useful in planning and conducting future population-based studies in a similar context elsewhere.

## 2. Study Description

The present analysis is based on a population-based nationwide health research project conducted in Bangladesh, a densely populated lower-middle-income country in South Asia, which aimed to evaluate the capacity of the primary health system to manage non-communicable diseases [22]. To provide a comprehensive and deeper analysis, this study examined both the supply-side factors (e.g., service delivery, allocation of resources, supplies, and logistics) and the demand-side factors (e.g., community readiness, context, and characteristics) of the health system in the context of non-communicable disease (NCD) prevention and management at the primary care level. A mixed-methods study design combining quantitative and qualitative approaches was employed (Table 1). From May to October 2021, data were collected from eight randomly selected sub-districts in four administrative districts—Cumilla, Jhenaidah, Rajshahi, and Sylhet—using a multi-stage cluster random sampling approach. The quantitative component was designed to assess the capacity of primary healthcare facilities to manage NCDs. Data were collected using a modified structured questionnaire adapted from the World Health Organization’s (WHO’s) Service Availability and Readiness Assessment (SARA) reference manual [23]. A total of 126 healthcare facilities were surveyed at various levels: public (e.g., Upazila Health Complex, Union Sub-Center, Union Health Center, and Community Clinic), private, and non-governmental organizations (NGOs) (clinics/hospitals) [24]. The heads of the respective healthcare facilities (managers or designated persons) were interviewed to collect health service data. Apart from that, household-level (*n* = 1645) data were collected from adult community members aged 18 years and above using standard survey questionnaires customized from existing literature to determine self-reported NCD prevalence, access to NCD care and services, health-seeking patterns, disease management practices, and the associated community’s characteristics and determinants. The qualitative component was designed to investigate the community characteristics and the circumstances influencing access to and the preference and willingness to receive NCD-related services and the barriers to and facilitating factors in organizing quality NCD care and services at the primary healthcare level [25,26]. Data were collected via 16 focused group discussions (FGDs) with community members, 15 in-depth interviews (IDIs) with local healthcare providers, and 14 key informant interviews (KIIs) with facility-based healthcare providers and managers (Table 1). A detailed description of the study design, sample size calculation, sampling strategies, and participant selection procedures were presented in our protocol study [22]. However, considering the available resources and time, we increased the household-level interviews from 1386, as proposed in the original protocol [22], to 1645 to achieve greater study power.

Village doctors are unqualified allopathic providers who usually received short training (from a few weeks to a few months) on common illnesses from unregulated and unregistered semi-private institutes that do not follow a standard curriculum [27].

Pharmacists are drugstore salespersons; most of them do not have formal training in dispensing, diagnosis, and treatment [27].

Upazila is an administrative structure that functions as a sub-unit of a district. The Upazila health complex is the first referral healthcare facility located in each Upazila.

## 3. The Data Collection Team and Training Workshop

A group of data enumerators—mostly graduates in anthropology and medicine—were employed to conduct the surveys and interviews. All of them had previously been involved in similar projects and had experience working on community-based research projects in the current data collection site. Certain hiring criteria were applied in the recruitment process, such as educational background, understanding of research methods and research instruments, motivation and willingness to work in rural settings, experience in technology-assisted (e.g., smartphone, personal data assistant) data collection techniques, and six months of availability to work. A two-week intensive training was provided via Zoom after they were recruited. The training sessions covered administration skills and demonstrated question/interview guidelines, including data collection methods and techniques and the use of Research Electronic Data Capture (REDCap) to collect and record the data [28]. All research instruments (e.g., questionnaires and interview guidelines) were piloted in the Jhenaidah and Rajshahi districts to assess their comprehensibility, conformability, and appropriateness. Necessary changes were made to the final research instruments based on the pilot study. The data collection procedure was supervised, and supportive supervision was provided by the first and third authors in regular group meetings and debriefing sessions via Zoom.

## 4. Ethical Considerations

The project received ethical approval from the Monash University Human Research Ethics Committee (Project ID: 27112) and the Bangladesh Medical Research Council (Ref: BMRC/NREC/2019-2022/270). Ethical values and standards were fully complied with throughout the research process. Participants’ confidentiality and anonymity were maintained at all stages of the study.

## 5. Challenges Faced and Mitigating Strategies Adopted

This section detailed the challenges faced and mitigating strategies adopted in this study. The key challenges faced and mitigating strategies adopted were categorized as follows: (i) challenges relating to the COVID-19 pandemic and mitigating strategies, (ii) challenges related to contextual factors and mitigating strategies, and (iii) challenges related to data quality and validity and mitigating strategies.

### 5.1. Challenges Relating to the COVID-19 Pandemic and Mitigating Strategies

Like it did many aspects of our everyday lives, the COVID-19 pandemic negatively affected population health research across the globe [14,15]. The COVID-19-related isolation measures and restrictions notably limited access to study sites due to insufficient transportation resulting from the nationwide lockdowns. In Bangladesh, the first COVID-19 confirmed case was recorded in March 2020 [29]. The Government of Bangladesh (GoB) took various measures to control the spread of infection. The vital controlling measures included several nationwide lockdowns, transportation restrictions, and social distancing measures between March and May 2020. At the beginning of May 2021, the Institute of Epidemiology, Disease Control and Research identified a new mutating variant of COVID-19 known as the Delta variant (or Indian variant). This time, the country underwent a new phase of nationwide lockdown, which was stricter than in 2020 [30,31,32]. The compulsory measures affected participatory research by restricting movement, minimizing person-to-person contact, enforcing social distancing, and preventing public gatherings. The newly imposed restrictions were stricter in districts adjacent to the India–Bangladesh border. Against this backdrop, the research team organized a consultation meeting with the field supervisor and data enumerators and reviewed the possible alternative strategies. Upon reaching a consensus, the team adopted a new strategy to conduct qualitative interviews with healthcare professionals instead of continuing with the household surveys, which most likely resulted in a more direct person-to-person contacts, particularly in a densely populated community. In the rural communities, it is likely that people believe that being interviewed would bring some sort of benefit (token money, project support) [33,34]. It is also likely that the curious masses around the interview place would cause a higher amount of person-to-person contact during the interviews in the community settings [21]. Interviewing the healthcare professionals required a minimal level of direct interaction, as the interviews were conducted in a private location at a convenient time (e.g., during the afternoon at their chambers). The interviews were conducted through telephone conversations rather than face-to-face meetings as originally planned. Simultaneously, the fieldwork in Rajshahi (a district adjacent to the Indian border) was temporally postponed due to a strict COVID-19 lockdown, so the team was moved to Jhenaidah. This adjustment to the field plan enabled researchers to continue their data collection activities. The team was sent back to Rajshahi in September 2021 after the COVID-19 restrictions in Bangladesh were withdrawn and everyday life was back to normal.

Furthermore, the questionnaires for both the facility and household surveys were thoroughly reviewed and simplified. The simplified, shorter version reduced the length of the interview process and provided more accurate information. We asked more closed-ended questions rather than open-ended questions, as they were more straightforward and could be answered more quickly. For example, instead of asking about the number of medicines taken for each treatment, a yes/no question was asked to reduce the length of the survey. Furthermore, the number of questions was reduced to ensure a good balance by merging a few of the queries and eliminating double-barreled questions.

### 5.2. Challenges Related to Contextual Factors and Mitigating Strategies

COVID-19-related infodemics as well as rumors and fear, misinformation, exclusion and neglect, stigma, and avoidance were common phenomena during the pandemic [35]. The widespread persistence of COVID-19-related infodemics and existing contextual factors caused community members to be hesitant to interact with data collection teams. Even though the data collection team consisted of graduates from universities adjacent to the study sites, many of the villages in the study were often unknown to them. Each district consists of several sub-districts and thousands of villages. As the villages were randomly selected, the data collection team was required to enter places completely unknown to them on many occasions. To allow for a quick, easy, and practical approach to the respondents in the local community, three supporting staff members from local communities were included in the team. The locally hired support staff facilitated greater acceptance of the data collection efforts by mobilizing the communities (e.g., interview participants, community leaders, and local healthcare providers) to participate in the study. Additionally, the locally hired staff helped familiarize the data collection team with the local customs, norms, and sentiments vital for cross-cultural interviews [36,37,38]. Being sensitive to local traditions, faiths, and religious beliefs and practices helped them to gain interviewees’ acceptance into the community and their consent. Adopting locally accepted dress and developing rapport with participants and community-level influentials (e.g., religious leaders, teachers, and locally elected personnel) facilitated engagement in intimate conversations [39,40,41]. Merging and routinely shuffling the teams as needed positively affected the fieldwork outcomes. For instance, more female interviewers were deployed in Sylhet and Cumilla, where religious beliefs and faith are stronger and more culturally sensitive [42] than in Rajshahi and Jhenaidah. During team formation, we considered applicants’ academic degrees and disciplines, prior experience, availability and commitments, proficiency in local languages, familiarity with the various data collection techniques and methods, communication skills, and openness to learning new methods and techniques. Combining social sciences and medical graduates within teams enabled them to access communities and healthcare facilities. Approaching the healthcare facility with a medical graduate taking the lead helped the teams gain greater access to the healthcare facility. Likewise, social science graduates effectively explained the objective of the research and gained acceptance in the communities and households. We found this strategy to be practical, quick, and cost-effective, and it resulted in quality data curation aligned with the field operation plan.

Repeated incidences of unexpected natural events such as heavy rainfall and flash floods affected data collection. In late July 2021, heavy rain and excessive water flow from the hills and upstream rivers triggered flash floods in Sylhet [43]. Sylhet is a low-lying flash flood-prone district. This district gets inundated almost every year due to the heavy rainfall and the onrush of water from the hilly/mountainous area of the Meghalaya and Assam states of India [44]. From late July to the beginning of August 2021, the field team experienced an incomprehensible flash flood while conducting the household survey [45]. The flash flood was sudden and overwhelmed the locality. Many families were severely affected by the floods and frequent rainfalls. The roads were underwater, and transportation was blocked. Moreover, many families became distressed over the shortage of food, clean water, and medical supplies. It was unfeasible to progress to the villages and approach the flood-affected inhabitants for an interview. In response to this unexpected event, the field plan was revised. As flash floods primarily affect rural communities, we employed the team in an urban area. By comparison, urban areas are resilient to climatic vulnerabilities and extreme events. This flexible field plan allowed the fieldwork to continue.

### 5.3. Challenges Related to Data Quality and Validity and Mitigating Strategies

The feasibility of conducting person-to-person interviews was limited due to COVID-19-related restrictions [46,47]. During the pandemic, the number of person-to-person contacts among the population, including the study participants, was significantly reduced, which might impact the data quality and validity [48]. To ensure the safety of the study participants, a range of alternative data collection methods and strategies were adopted in the pandemic context; however, the issues of data quality and validity remain a concern [15]. Data were collected using a mixed-methods approach (combining quantitative and qualitative methods), as our study addressed complex and multiple perspectives of the primary healthcare system in managing non-communicable diseases in Bangladesh [22]. This approach allowed us to involve participants from various backgrounds, roles, sites, and perspectives (Table 1). Multiple data collection tools/methods were used to collect data from various strata (e.g., age, sex, occupation, geographical location) that helped us to gather quality and reliable data [49]. However, several factors, such as availability of time, finance, and needed skillsets were crucial to adopting this approach [50]. Carefully selecting team members from diverse educational backgrounds and experience provided the skillset required for a mixed-methods approach [50,51]. The team members, including data collectors and a field supervisor, were hired from participating study sites (e.g., graduates from universities adjacent to field sites). This hiring strategy maximized the use of allocated time and resources by reducing the number of intra-site visits (within the study sites) and minimized the frequency of face-to-face contact with study participants. Regarding generating quantitative data at the healthcare facilities, a combined approach of interviewing and observations was adopted. The observation method was considered a useful complementary approach to minimize the person-to-person contact with healthcare workers when collecting certain medical and infrastructure information, which could be a viable strategy to generate quality and validity in the COVID-19 context [52]. Curation of quality and valid textual data (e.g., IDI, KII, and FGD) via online means/telephone interviews remain an ongoing debate due to the suspected losses of contextual and non-verbal expressions of the participants, lack of rapport, and proving and noting visual cues, which are important elements of interpreting the results [53,54]. Amid the ongoing debates, the existing evidence supports the views of using internet-based online platforms as viable alternative methods for generating textual data upon acknowledging associated limitations and careful consideration of any technical difficulties [55,56]. However, the adaptation of these alternative methods was considered unsuitable because the participants had no or insufficient access to internet-based resources and opportunities, a low level of familiarization/ability, and a lack of comfortability with their usage. Therefore, we decided to conduct interviews using traditional methods (face-to-face interviews). In this context, the quality and validity of the data collected were maintained by applying a stepwise procedure before, during, and after interviews. At the pre-interview step, the research team members independently reviewed the literature to develop topic guides/items, sought and incorporated experts’ views from respective study domains, and piloted topic guides in multiple settings adjacent to the original field sites, as detailed in our previous papers [25,26]. Required modifications were made to format and validate data collection instruments. A field supervisor with extensive experience in the implementation of population-based participatory research in similar settings was engaged to promote and coordinate data collection activities at study sites. During the interviews, the field supervisor supervised and monitored the data collection activities, supported coordination with healthcare facility managers and local-level staff to access the field sites, organized regular meetings with data enumerators and research team members, shared their independent reviews, and organized debriefing to maintain standard procedures for collecting data [57]. In the post-interview stage, each author dependently reviewed and analyzed data to increase the validation of the information gathered. Formal discussions among the team members were conducted to reach a consensus when disagreements arose. It is assumed that the authors’ careful consideration of multi-step procedures in developing the research instruments and their successful implementation enabled the collection of quality data.

## 6. Conclusions

This paper describes a subjective reflective account of challenges faced and mitigating strategies used in conducting a population-based primary research project during the COVID-19 pandemic in Bangladesh. We conclude that a proper understanding of pandemic-related challenges within the context is crucial to designing and implementing population-based research. Our experiences suggest that the challenges faced during the project were multi-faceted, including COVID-19 pandemic-related controlling measures, contextual challenges, data quality, and validity. A range of adaptive strategies, including designing contextually adapted research methods and instruments, optimum use of locally available resources (e.g., hiring field enumerators from study sites and involving local supervisors), and situation-driven actions (the modification of data collection tools and operation plans, team building, and responding to local values and culture) were effective strategies for conducting a population-based health research project during the uncertain times. The experience of our study may be useful to those who are planning to conduct future population-based health research locally and in a similar context elsewhere.

## Figures and Tables

**Table 1 ijerph-20-05629-t001:** Data collection processes and the selection of study participants.

Methods	Study Sites (Districts)			
	Cumilla	Jhenaidah	Rajshahi	Sylhet
Household survey (*n* = 1645)	Households (*n* = 675)	Households (*n* = 214)	Households (*n* = 316)	Households (*n* = 440)
Facility survey (*n* = 126)	Healthcare facility (*n* = 56)	Healthcare facility (*n* = 17)	Healthcare facility (*n* = 28)	Healthcare facility (*n* = 25)
In-depth interviews (IDIs) [*n* = 15]	IDI1: With front-line health staff [health assistant/family welfare visitor/sub-assistant community medical officer] (*n* = 2); IDI2: With private vendor/pharmacist/traditional provider [village doctor, faith healers, Kabiraj] (*n* = 2) [in each district (*n* = 3–4)]			
Focus group discussions (FGDs) [*n* = 16]	FGD1: With the community in rural areas [people with NCDs] (*n* = 2); FGD2: With the community in urban areas [people with NCDs] (*n* = 2) [in each district (*n* = 4)]			
Key informant interviews (KIIs) [*n* = 14]	KII1: With an Upazila health and family planning officer/medical officer/residential medical officer (*n* = 8) [in each district (*n* = 2)]; KII9–10: With a district health manager [civil surgeon] (*n* = 2); KII11–12: Policy a planner/independent consultant/specialist (*n* = 2); KII13–14: Private doctor/NGO workers at districts (*n* = 2)			

FGDs: Focused Group Discussions, IDIs: In-depth Interviews, KIIs: Key Informant Interviews, NCDs: Non-communicable Diseases, Kabiraj is mostly a self-trained traditional healer whose practices are based on diets, herbs, creepers, and exercise [27].

## Data Availability

Not applicable.

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
