# Peer review of "Challenges and Strategies in Conducting Population Health Research during the COVID-19 Pandemic: Experience from a Nationwide Mixed-Methods Study in Bangladesh"

_ijerph, 2023, doi:10.3390/ijerph20095629_

Round 1

Reviewer 1 Report

#1. section 2. NCD >> = non-communicable diseases?

#2. ref-24 >> Was it https://doi.org/10.21203/rs.3.rs-1664974/v1 ? if yes, suggest to add the DOI so the readers can find this references.

#3. The study aimed to “account of the challenges faced by researchers conducting nationwide population health research during the COVID-19 pandemic and the mitigating strategies”. However, in section 5 “Challenges faced and mitigating strategies adopted”, the 2nd subsection “(ii) Challenges related to contextual factors and mitigating strategies” did not clearly report any COVID related issues.

#4. This manuscript reported the experience from a nationwide mixed-methods study in Bangladesh. Some terminologies [such as Kabiraj and Upazila] in this manuscript were specific to Bangladesh and/or South Asia and not familiar to the international audience, so adding some explanations or references might be helpful.

Author Response

Comment: #1. section 2. NCD >> = non-communicable diseases?

 Response: Thank you for pointing this out. We have used the non-communicable disease to elaborate the NCD at the first use (line 93: page:2)

Comment: #2. ref-24 >> Was it https://doi.org/10.21203/rs.3.rs-1664974/v1 ? if yes, suggest to add the DOI so the readers can find this references.

 Response: We have added the DOI in the cited reference in the revised version.

Comment: #3. The study aimed to “account of the challenges faced by researchers conducting nationwide population health research during the COVID-19 pandemic and the mitigating strategies”. However, in section 5 “Challenges faced and mitigating strategies adopted”, the 2nd subsection “(ii) Challenges related to contextual factors and mitigating strategies” did not clearly report any COVID related issues.

 Response: Thank you for this comment. We have added the following texts to address this point. (line: 195-200, page: 7)

‘COVID-19-related infodemics such as rumor and fear, misinformation, exclusion and neglects, stigma, and avoidance had been common phenomena during the pandemic (2, 35). The widespread persistence of COVID-19-related infodemic and existing contextual factors caused community members hesitant to interact with data collection teams.’

Comment: #4. This manuscript reported the experience from a nationwide mixed-methods study in Bangladesh. Some terminologies [such as Kabiraj and Upazila] in this manuscript were specific to Bangladesh and/or South Asia and not familiar to the international audience, so adding some explanations or references might be helpful.

Response: Thank you for pointing out these issues. We have added the following texts to clarify these terminologies. Please see the footnote of table 1. (lines: 124-134, pages: 3-4).

FGDs: Focused Group Discussions, IDIs: In-depth Interviews, KIIs: Key Informant Interviews, NCDs: Non-communicable Diseases.

Kabiraj is mostly a self-trained traditional healer whose practices are based on diets, herbs, creepers, and exercise. (27)

Village doctors are unqualified allopathic providers who usually received short training on common illnesses (from a few weeks to a few months) from unregulated and unregistered semi-private institutes which do not follow a standard curriculum. (27)

Pharmacists are drugstore salespersons; most of them do not have formal training in dispensing, diagnosis, and treatment. (27)

Upazila is an administrative structure that functions as a sub-unit of a district. Upazila health complex is the first referral healthcare facility located in each Upazila.

Reviewer 2 Report

The current manuscript describes the specific measures for populational research during COVID-19 pandemic. The article is more descriptive than reflective. Experience of authors is valuable. Still, it requires consideration and reflection to the measures they took including pros and cons. How effective was the strategy (e.g. mixed-methods study)? As face-to-face interview was restricted, interviewing professionals were done instead of interviewing household (section 5i). Would there be a drawback? Any improvement could be made? To shed light on the best measures for populational research during pandemic situation, a more solid model could be given. Point-to-point format discussion could be done (e.g. one challenge and one measure).

Author Response

Comment: The current manuscript describes the specific measures for populational research during COVID-19 pandemic. The article is more descriptive than reflective. Experience of authors is valuable. Still, it requires consideration and reflection to the measures they took including pros and cons. How effective was the strategy (e.g. mixed-methods study)? As face-to-face interview was restricted, interviewing professionals were done instead of interviewing household (section 5i). Would there be a drawback? Any improvement could be made? To shed light on the best measures for populational research during pandemic situation, a more solid model could be given. Point-to-point format discussion could be done (e.g. one challenge and one measure).

Response: We appreciate this insightful comment. This paper mainly reflected a subjective account of challenges and adopting mitigating strategies in the context of implementing population-level health research during the COVID-19 pandemic. We have clarified this information in the entire manuscript. There was limited scope to present a critical reflective overview of methods and approaches adopted in this study due to the focus on the subjective experience of the authors/team members in line with some contextual factors. However, we have added texts in some places that better reflect what overcoming strategies were taken under which context and how these worked. The following texts were added (lines: 263-277, page: 6). Finally, each category of challenges was first defined which was ended following a discussion of strategies taken in the existing version.

‘Data were collected using a mixed-method approach (combining quantitative and qualitative methods) as such our study addressed complex and multiple perspectives of the primary healthcare system in managing non-communicable diseases in Bangladesh (22). This approach allowed us to involve participants from various backgrounds, roles, sites, and perspectives (Table 1). Multiple data collection tools/methods were used to collect data from various strata (e.g., age, sex, occupation, geographical locations) that helped to gather quality and reliable data (49). However, several factors such as availability of time and finance, and needed skill sets were crucial for adopting this approach (50). A careful consideration for selecting team members from diverse educational backgrounds and experience provided the skill set required for a mixed-method approach (50, 51). The team members including data collectors and a field supervisor were hired from participating study sites (e.g. graduates from universities adjacent to field sites). This hiring strategy maximized the use of allocated time and resources by reducing the number of intra-sites visits (within the study sites), and minimized the frequency of face-to-face contact with study participants.’

Round 2

Reviewer 1 Report

In response to my previous comments “. ref-24 >> Was it https://doi.org/10.21203/rs.3.rs-1664974/v1 ? if yes, suggest to add the DOI so the readers can find this references.”, the authors replied “Response: We have added the DOI in the cited reference in the revised version.”. >> The reviewer can’t find it in the revised manuscript.

Author Response

Response to reviewer

Comment: In response to my previous comments “. ref-24 >> Was it https://doi.org/10.21203/rs.3.rs-1664974/v1? if yes, suggest to add the DOI so the readers can find this references.”, the authors replied “Response: We have added the DOI in the cited reference in the revised version.”. >> The reviewer can’t find it in the revised manuscript.?

Response: Thank you, the reviewer, for bringing up this issue.

Citations 24 and 25 are based on the same data. However, citation 24 (Cited as Kabir A, Karim MN, Billah B. Readiness of primary healthcare facilities for non-communicable diseases in Bangladesh. Research Square; 2022.) is the pre-print version of citation 25 which has been published in BMC Primary Care (cited as Kabir A, Karim MN, Billah B. The capacity of primary healthcare facilities in Bangladesh to prevent and control non-communicable diseases. BMC Primary Care. 2023;24(1):60.). Therefore, we have removed citation 24 in this version.

Reviewer 2 Report

No further revision is required. Authors have declared the focus of the current manuscript is to share subjective experience. 

Author Response

Thank for your comment.